# CROSS-SPECIES PROTEIN INTERACTION PREDICTION BY HIERARCHICAL CONTRASTIVE PRETRAINING

## ABSTRACT

Protein-protein interaction prediction is fundamental for understanding cellular processes, yet most existing approaches struggle with both intra-species accuracy and cross-species generalization. We present HIPPO, a hierarchical contrastive framework for protein-protein interaction prediction across organisms. HIPPO integrates amino acid sequences, biological hierarchies, and functional annotations into a unified representation learning objective. By aligning proteins not only at the sequence level but also according to their hierarchical relationships, HIPPO enforces embeddings that reflect the multi-level organization of protein functions. This structured supervision enables more accurate predictions within species while also facilitating transfer to unseen proteins and species. To capture global network context, protein embeddings are propagated through interaction graphs using graph neural architectures. Experiments on benchmark datasets demonstrate that HIPPO achieves consistent state-of-the-art performance, with substantial improvements in both intra-species and cross-species prediction. Crucially, extensive interpretability analyses reveal that hierarchical supervision highlights conserved motifs, binding residues, and post-translational modification regions, yielding biologically grounded interpretability and improving the reliability of protein interaction discovery.

## 1 INTRODUCTION

Protein–protein interactions (PPIs) are central to virtually all cellular processes and play a critical role in elucidating molecular mechanisms, disease pathways, and therapeutic targets (Vidal et al., 2011; Stumpf et al., 2008). Accurate mapping of PPIs is indispensable for understanding how proteins coordinate to regulate biological systems (Stelzl et al., 2005; Rual et al., 2005). Experimental techniques such as yeast two-hybrid assays, co-immunoprecipitation, pulldown experiments, chemical cross linking, and proximity-based labeling have provided valuable insights (Brückner et al., 2009a; Kaboord & Perr, 2008a; Tan et al., 2016a; Cho et al., 2020a). However, these methods are costly, labor intensive, and limited in scalability. In well studied species such as human and yeast, lots of efforts have yielded valuable interactome data (Rolland et al., 2014; Yu et al., 2008). By contrast, many organisms remain under characterized, with only sparse PPI data available (Consortium, 2011; Mosca et al., 2013).

Many computational approaches struggle when applied to less-studied organisms, primarily because training datasets of PPIs must be sufficiently large to provide representative interaction samples. A major barrier to bridging the species gap is that there are often too few experimentally verified interactions to train PPI inference models for these organisms (Mosca et al., 2013; Consortium, 2011). For computational models to effectively address this challenge, they must be capable of making accurate PPI inferences on less-studied organisms while leveraging data from well-characterized species during training (Sharan et al., 2005; Kuchaiev et al., 2010).

Based on xxxx.Cross-species inference provides a promising solution, as it enables the prediction of intraspecies PPI networks for a target organism using training data from another, typically well-studied, species. This setting requires models to perform accurate out-of-distribution (OOD) inference, transferring knowledge across evolutionary boundaries. However, existing PPI inference methods have historically failed to generalize in this scenario, producing inaccurate predictions when both interacting proteins belong to OOD distributions (**?**). Based on prior work on network alignment and cross-species function transfer (Sharan et al., 2005; Kuchaiev et al., 2010), cross-species

inference provides a promising solution, as it enables the prediction of intra species PPI networks for a target organism using training data from another, typically well-studied, species. This setting requires models to perform accurate out-of-distribution (OOD) inference, transferring knowledge across evolutionary boundaries (Mosca et al., 2013; Littmann et al., 2021). However, existing PPI inference methods have historically failed to generalize in this scenario, producing inaccurate predictions when both interacting proteins belong to OOD distributions (Bileschi et al., 2022; Rolland et al., 2014).

In this paper, we introduce HIPPO, a hierarchical multimodal framework that integrates sequence, annotation, and hierarchical supervision into a unified contrastive learning objective. HIPPO not only outperforms within species but also substantially enhances cross-species transfer, especially for Protein pairs with less similarity. Importantly, it offers interpretability by revealing conserved motifs and functional sites captured by attention analysis.

Our main contributions are:

- We introduce HIPPO, a hierarchical contrastive framework that unifies sequence, annotation, and biological hierarchy relationship for protein–protein interaction prediction across species.
- We demonstrate that hierarchical supervision substantially improves predictive performance in both intra-species and cross-species settings, particularly on challenging cases involving unseen proteins and species.
- We conduct extensive interpretability analyses and find that HIPPO highlights conserved motifs, binding residues, and post-translational modification regions, thereby providing biologically meaningful explanations for model predictions.

## 2 RELATED WORK

### 2.1 CROSS SPECIES PREDICTION

Cross-species function prediction aims to transfer biological knowledge across organisms by exploiting evolutionary conservation. Early approaches primarily relied on sequence homology, under the assumption that orthologous genes share similar functions **?**. While effective in closely related species, the accuracy of homology-based transfer degrades with evolutionary distance and fails to capture species-specific functions. To overcome these limitations, network-based approaches have been developed to align protein–protein interaction networks or integrate multi-omics evidence across organisms Sharan et al. (2005); Kuchaiev et al. (2010). More recently, representation learning methods have extended cross-species prediction by incorporating diverse biological signals. Large-scale protein language models provide universal embeddings that generalize across species and improve annotation in less-studied organisms Bileschi et al. (2022); Littmann et al. (2021). In parallel, deep metric learning frameworks such as INTREPPPID introduce orthology-informed quintuplet networks for cross-species protein–protein interaction prediction, combining sequence and interaction features to improve out-of-distribution generalization. Together, these advances expand the coverage of functional annotation beyond model organisms and facilitate applications in translational medicine, agriculture, and synthetic biology.

### 2.2 PROTEIN-PROTEIN INTERACTION PREDICTION

Protein–protein interactions are fundamental molecular events in vivo and represent key targets for therapeutic interventions. Experimental assays such as yeast two-hybrid (Brückner et al., 2009b), co-immunoprecipitation (Kaboord & Perr, 2008b), pull-down (Aronheim et al., 1997), cross-linking (Tan et al., 2016b), and proximity labeling (Cho et al., 2020b) have been widely used to detect and characterize interactions. However, the high cost and limited throughput of these techniques make it infeasible to comprehensively map unknown interactions. As a result, computational approaches have emerged as efficient and scalable alternatives for predicting protein–protein interactions and mapping the human interactome.

With recent advances in artificial intelligence, PPI prediction has shifted from traditional machine learning to deep learning frameworks. Current methods largely focus on learning protein represen-

tations from sequence and structure. Sequence-based approaches remain the most widely used for functional prediction, with architectures including convolutional neural networks (Shanehsazzadeh et al., 2011), recurrent networks such as LSTMs (Alley et al., 2019), Transformers (Elnaggar et al., 2022), and dilated residual networks (Rao et al., 2019).

## 2.3 INTERPRETABILITY ANALYSIS

Interpretability is increasingly recognized as a critical requirement for biological prediction models, particularly when aiming to generate mechanistic insights or guide downstream experiments. Early interpretability efforts relied on post-hoc attribution methods, such as saliency maps or attention scores, to highlight influential residues or sequence motifs (Sundararajan et al., 2017; Vig et al., 2020). In the context of protein research, attention-based models have revealed functional sites, conserved domains, and binding interfaces (Rao et al., 2021; Rives et al., 2021). For graph-based models, interpretability techniques such as node/edge masking, gradient-based relevance propagation, and subgraph extraction have been employed to identify key residues and structural motifs (Ying et al., 2019; Vu et al., 2023). Furthermore, biology-specific interpretability frameworks have emerged, including methods that correlate learned embeddings with known ontologies, Pfam families, or Gene Ontology terms (Littmann et al., 2021; Bepler & Berger, 2021). These methods not only improve trust in computational predictions but also enable hypothesis generation, such as identifying putative binding sites or uncovering conserved substructures across species.

## 3 METHOD

### 3.1 HIERARCHICAL PRETRAINING METHODS

To enable effective downstream prediction of protein-protein interactions, we introduce a pretraining framework that integrates hierarchical relationships and functional annotations into protein sequence representations. The framework is composed of two complementary components: hierarchical contrastive learning and multimodal sequence and annotation alignment.

**Hierarchical Contrastive Learning for Hierarchical Labels**   Protein functions and structures are naturally organized into hierarchical categories such as clans, families, and domains. To reflect this structure, we adapt the Hierarchical Multi-label Constraint Enforcing Contrastive Loss (HiMul-ConE) (**??**), which enforces consistency across hierarchical levels.

Given a set of levels $L$, the objective ensures that ancestor labels (e.g., clan) maintain higher confidence than their descendants (e.g., family). Formally, the hierarchical contrastive loss is

$$\mathcal{L}_{\mathrm{HC}} = \sum_{l \in L} \frac{1}{|L|} \sum_{i \in I} \frac{-\lambda_l}{|P(i)|} \sum_{p_l \in P_l} \max \left( L^{\mathrm{pair}}(i, p_l^i), L_{\max}^{\mathrm{pair}}(l-1) \right), \tag{1}$$

where $f_i$ and $f_p^l$ are protein feature representations at level $l$, $\lambda_l$ is a weighting factor, and $L^{\mathrm{pair}}(i, p_l^i)$ is a contrastive pair loss:

$$L^{\mathrm{pair}}(i, p_l^i) = \log \frac{\exp(f_i \cdot f_p^l / \tau)}{\sum_{a \in A \setminus i} \exp(f_i \cdot f_a / \tau)}. \tag{2}$$

This design encourages embeddings to respect the hierarchical organization of protein functions.

**Multimodal Sequence-Annotation Alignment**   To incorporate curated biological knowledge, we align protein sequence embeddings with their corresponding functional annotation embeddings. Following ProteinBERT (Brandes et al., 2022), we encode sequences using a protein language model (PLM), producing sequence embeddings $\{\mathbf{z}_i^S\}_{i=1}^N$, and annotations using an annotation language model (ALM), producing embeddings $\{\mathbf{z}_i^T\}_{i=1}^N$.

We first employ a symmetric InfoNCE loss (van den Oord et al., 2018) to maximize the agreement between matching sequence–annotation pairs:

$$\mathcal{L}_{\mathrm{SAC}} = -\frac{1}{2N} \sum_{i=1}^N \left( \log \frac{\exp(\mathbf{z}_i^S \cdot \mathbf{z}_i^A / \tau)}{\sum_{j=1}^N \exp(\mathbf{z}_i^S \cdot \mathbf{z}_j^T / \tau)} + \log \frac{\exp(\mathbf{z}_i^A \cdot \mathbf{z}_i^T / \tau)}{\sum_{j=1}^N \exp(\mathbf{z}_j^A \cdot \mathbf{z}_i^T / \tau)} \right). \tag{3}$$

To further distinguish true matches from mismatches, we introduce a Sequence–Annotation Matching (SAM) loss based on focal loss:

$$\mathcal{L}_{\text{SAM}} = -\frac{1}{N}\sum_{i=1}^{N}\left(\alpha(1-p_i)^\gamma \log(p_i)\cdot y_i^{\text{SAM}} + (1-\alpha)p_i^\gamma \log(1-p_i)\cdot(1-y_i^{\text{SAM}})\right), \quad (4)$$

where $p_i$ is the predicted probability of a valid sequence–annotation match and $y_i^{\text{SAM}}$ is the ground truth indicator.

The final pre-training objective combines hierarchical supervision and multimodal alignment:

$$\min_{\theta}\left[\mathcal{L}_{\text{HC}} + \mathcal{L}_{\text{SAC}} + \mathcal{L}_{\text{SAM}}\right], \quad (5)$$

where $\theta$ includes all parameters of the PLM, ALM, and projection heads. This objective encourages protein representations that simultaneously respect biological hierarchies and align with functional annotations.

## 4 EXPERIMENT

### 4.1 EXPERIMENTAL SETUP

In this section, we first describe the experimental setup for pretraining and evaluation. We then present results on protein–protein interaction (PPI) prediction across both intra-species and cross-species settings, followed by interpretability analyses. Additional experiments and ablation studies are provided in Appendix A2-3.

**Pretraining datasets** We use the 440K Swiss-Prot protein sequence database (Bairoch & Apweiler, 2000) for pretraining. This database provides high quality, manually curated protein sequences and annotations. We map the proteins to the Pfam database to obtain hierarchical labels, including protein families and clans. Annotations from Swiss-Prot are derived from the keywords section, which incorporates controlled vocabulary terms manually curated to cover Gene Ontology (GO) terms, disease associations, protein domains, ligands, and post-translational modifications (PTMs) (The UniProt Consortium, 2025).

**Protein-Protein Interaction prediction** We seeks to classify 7 types of PPI pairs within the interactome(Lv et al., 2021) across intra- and cross- species mode. PPI datasets comes from STRING database(Szklarczyk et al., 2021; 2025), where human dataset include SHS148k and SHS27k. SHS27k comprises 63,408 interactions among 1,690 proteins, while SHS148k includes 36,902 interactions among 5,189 proteinsLv et al. (2021). Besides human, we adopt interactome datasets from Escherichia coli, Saccharomyces cerevisiae (yeast), Mus musculus (mouse), Caenorhabditis elegans, Arabidopsis thaliana, and Drosophila melanogaster.

**Dataset splits** For intra-species PPI predictionwe follow Breadth-First Search (BFS), and Depth-First Search (DFS) method in Lv et al. (2021) to ensure that the test set contains PPI pairs containing proteins with less similarity to the training set(Lv et al., 2021; Wang et al., 2023). For cross species prediction, training set comes from human, and test set contains PPI from different organisms(Wang et al., 2023), where proteins are different origins. To better evaluate generalizability on intra-speices dataset, we further divide the test data into two subsets based on whether or not two proteins have been seen in the training data, including(1) Easy: either one proteins has been seen; and (2) Hard: neither one has seen.

**Baselines** Following prior work (Hu et al., 2023), we compare our framework with a diverse set of protein representation learning methods. Specifically, we include graph neural network–based PPI predictors (GNN-PPI (Lv et al., 2021), two large-scale pretrained protein language models (Protein-BERT (Brandes et al., 2022) and ESM2 (Lin et al., 2023)), one sequence-based deep learning method (PIPR (Chen et al., 2019)). Moreover, we include one model reported to deal with cross species PPI prediction(Wang et al., 2023). These baselines cover a broad range of modeling paradigms, including sequence-driven approaches, pretrained language models, and graph-based architectures.

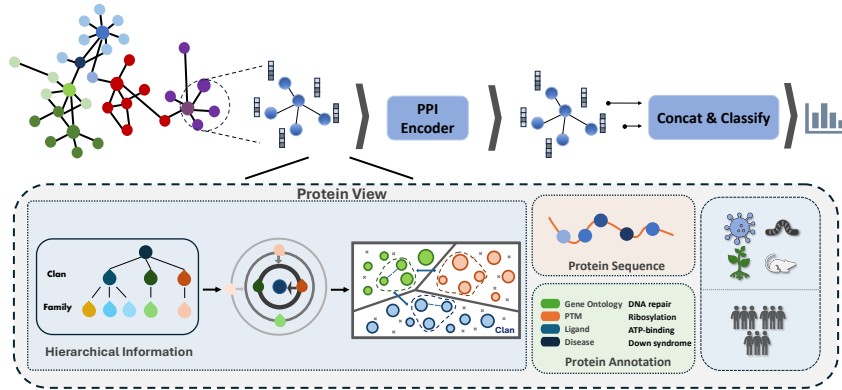

Figure 1: **Overview of HIPPO.** The framework integrates protein sequences, hierarchical annotations, and PPI networks to predict interactions.

### 3.2 NETWORK CONSTRUCTION FOR PROTEIN PAIRS

We represent protein-protein interaction (PPI) complexes as a labeled graph. Given a protein set

$$\mathcal{P} = \{p_0, p_1, \ldots, p_n\}$$

and the corresponding set of protein interactions

$$\mathcal{X} = \{x_{ij} = \{p_i, p_j\} \mid i \neq j, p_i, p_j \in \mathcal{P}\},$$

we define a label space

$$\mathcal{L} = \{l_0, l_1, \ldots, l_6\},$$

where each interaction $x_{ij}$ is associated with a multi-label set $y_{ij} \subseteq \mathcal{L}$. The dataset can then be written as

$$\mathcal{D} = \{(x_{ij}, y_{ij}) \mid x_{ij} \in \mathcal{X}\},$$

and naturally represented as a graph $\mathcal{G} = (\mathcal{P}, \mathcal{X})$ with proteins as nodes and interactions as labeled edges.

To incorporate functional and evolutionary context, each protein node is encoded into a hierarchical embedding informed by family- and clan-level annotations. These embeddings capture evolutionary lineage and structural similarity, providing richer context for modeling interaction mechanisms.

The PPI prediction task is formulated as learning a mapping

$$\mathcal{F} : x_{ij} \rightarrow \hat{y}_{ij},$$

using a training set $\mathcal{X}_{\text{train}} \subseteq \mathcal{X}$ and evaluating on a disjoint test set $\mathcal{X}_{\text{test}}$, with $\mathcal{X}_{\text{train}} \cup \mathcal{X}_{\text{test}} = \mathcal{X}$.

We adopt the Graph Isomorphism Network (GIN) (Xu et al., 2019) as the backbone encoder. GIN aggregates neighborhood information to produce discriminative node embeddings. For a protein pair $x_{ij}$, the node embeddings $g_{p_i}$ and $g_{p_j}$ are combined via dot product and passed through a fully connected (FC) layer for prediction. We train the model with a binary cross-entropy loss over all interaction types:

$$L = \sum_{k=0}^{n} \sum_{x_{ij} \in \mathcal{X}_{\text{train}}} \left( -y_{ij}^k \log \hat{y}_{ij}^k - (1 - y_{ij}^k) \log(1 - \hat{y}_{ij}^k) \right),$$

where $y_{ij}^k$ and $\hat{y}_{ij}^k$ denote the ground truth and predicted probability of the $k$-th interaction type for protein pair $x_{ij}$.

**Training** For PPI prediction, the model is trained for 100 epochs , covering both intra-species and cross-species settings. To ensure fair comparison with baselines, all models are trained under the same optimization protocol. For pretraining, we adopt two complementary tracks depending on the type of labels: 1. Hierarchical labels derived from Pfam families and clans are used as supervisory signals, guiding the model to learn biologically structured representations that reflect

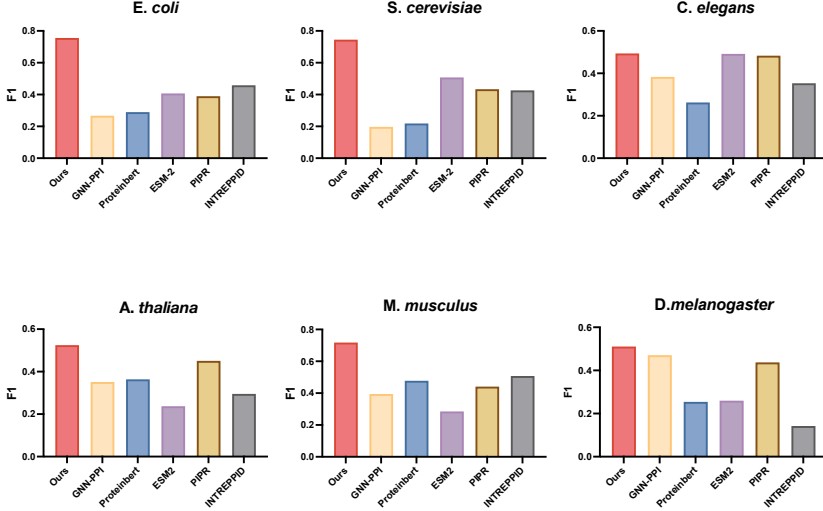

Figure 2: **Cross-species PPI prediction.** HIPPO outperforms or matches baselines (GNN-PPI, ProteinBERT, ESM-2, PIPR, INTREPPPID) across six organisms.

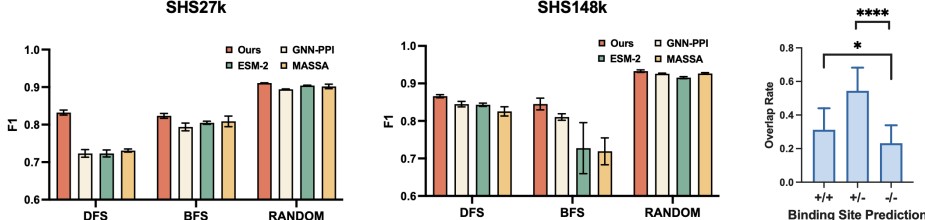

Figure 3: **Performance on human PPI datasets.** HIPPO achieves higher F1 scores than GNN-PPI, ESM-2, and MASSA under DFS, BFS, and random sampling.

evolutionary and functional hierarchies. 2. Non-hierarchical functional descriptors such as UniProt keywords are incorporated through an unsupervised contrastive objective (Chen et al., 2020; He et al., 2020), encouraging proteins with similar descriptors to cluster in the embedding space. The model is pretrained on Swiss-Prot for 100 epochs. All these models are trained on 8 GTX4090 GPUs. Detailed hyperparameter settings can be found in Appendix C.2.

**Evaluation** For PPI prediction, we evaluate the performance with the micro F1 values, which is common used in the PPI prediction task(Lv et al., 2021; Zhang et al., 2022; Song et al., 2022). Models with the best performance on validation sets are selected for evaluation.

## 4.2 RESULTS

**HIPPO outperforms all baselines on Intra- and Cross-Species PPI Prediction** As summarized in Fig.2, our model consistently outperforms existing methods, including GNN-PPI, ProteinBERT, ESM-2, PIPR, and INTREPPID, across all 6 cross-species datasets. The proposed approach achieves the highest F1 scores in each species, demonstrating its robustness and superior predictive power in challenging cross-species settings. These results highlight the capacity of our hierarchical, multimodal framework to learn transferable protein representations that generalize well to novel proteomes. For intra-species prediciton, HIPPO also achieves the best average performance according to Fig.2A. On SHS27k dataset, the improvement is most pronounced under the DFS split, where our model substantially outperforms GNN-PPI.On SHS148k dataset, our method continues to yield

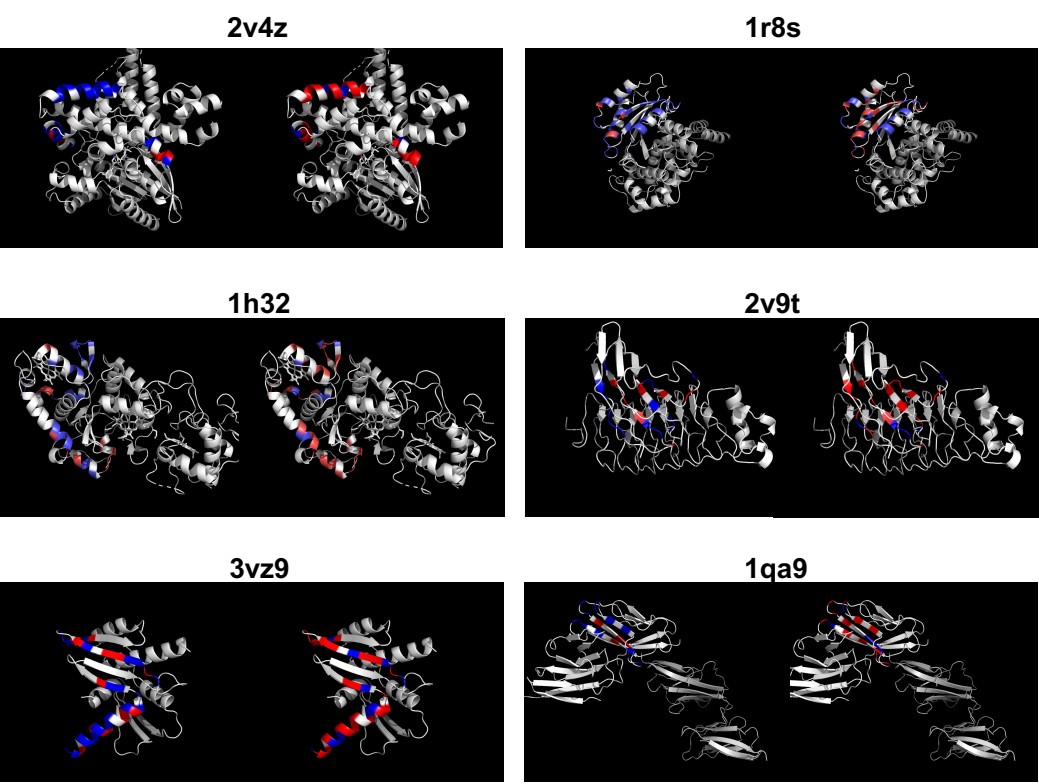

Figure 4: **Interpretability of binding site prediction.** Visualization on six proteins shows that hierarchical (HRC) pretraining improves accuracy, with correct sites in red and errors in blue.

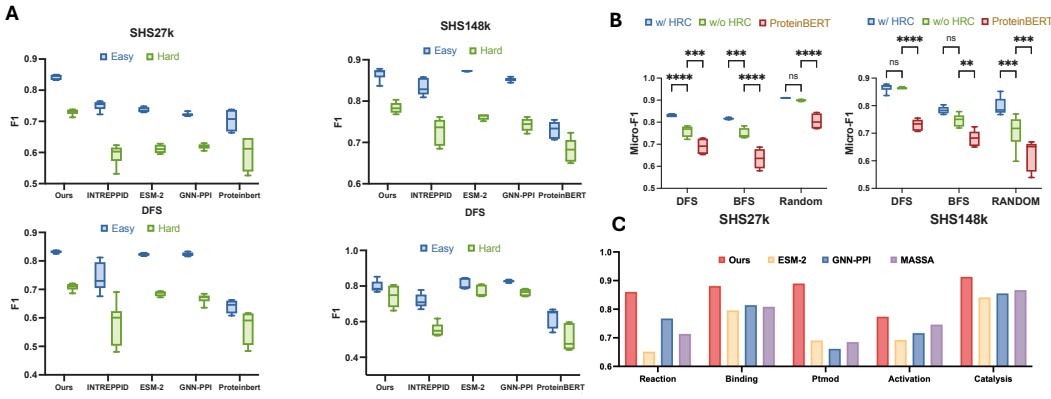

Figure 5: Sample figure caption.

consistent gains, surpassing GNN-PPI under both DFS and BFS splits. In contrast, performance differences narrow under the random split.

**HIPPO demonstrates superior performance in predicting unseen protein pairs.** We stratified protein-protein pairs into "easy" and "hard" categories: in the former, at least one interacting protein is seen during training, while in the latter, both proteins are unseen. As shown in Fig. 5B, all models achieve comparable F1 scores on easy pairs across both datasets and split strategies, indicating similar predictive capacity when at least partial training information is available. However, the differences between models become much more pronounced on hard pairs. While all methods ex-

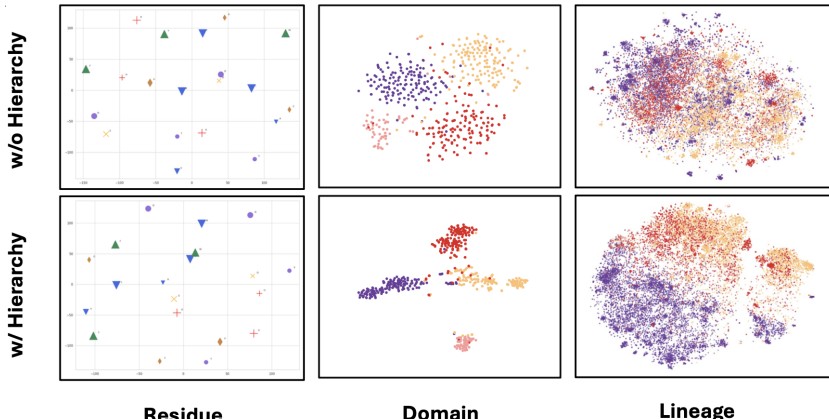

Figure 6: **Performance on interaction types and difficulty levels.** (A) Box plots on SHS27k comparing easy and hard subsets under DFS and BFS, showing consistent gains of our method. (B) Micro-F1 scores on SHS27k and SHS148k under DFS, BFS, and random sampling, where hierarchical(HRC) pretraining outperforms both without HRC and ProteinBERT, with significance indicated by ns, **, ***, and ****. (C) F1 scores of four models(Ours, ESM2, GNNPPI, MASSA) on five interaction types of SHS27k under DFS sampling.

perience a performance drop in this scenario, our model consistently outperforms the baselines and exhibits lower prediction variance. This advantage is particularly notable on the SHS27k dataset, where the gap between our method and others widens substantially for hard cases.

**Attention based Interpretability Reveals Functional Binding Site** We utilized attention weights extracted from the model to identify functional three dimensional binding site positions within protein–protein interaction complexes. According to Fig. 4, regions shown in red correspond to correctly predicted binding sites, while those in blue indicate missed sites. Incorporating hierarchical contrastive learning leads to a substantially higher overlap rate for binding site prediction in this complex compared to training without hierarchical supervision, highlighting HIPPO's ability to precisely capture interaction interfaces.

As shown in Fig.3, we further evaluated binding site prediction across a broader set of 35 protein complexes using three model configurations: hierarchical information with flat annotations, hierarchical information alone, and no annotations. Models with hierarchical supervision consistently outperformed the annotation-free model, confirming the importance of structured information for interface detection. Interestingly, adding flat annotations together with hierarchical labels slightly reduced performance relative to using hierarchical information alone, suggesting that non-hierarchical labels may introduce noise and weaken the model's ability to localize functional regions.

### 4.3 ABLATION STUDIES

As shown in Fig. 5B and Tab.1, removing hierarchical relationships among protein properties leads to a clear reduction in predictive accuracy across all intra-species datasets. The decline is most evident on challenging test pairs where both interacting proteins are unseen during training, indicating that hierarchical modeling is particularly important for generalizing to novel or under-annotated proteins.

To further assess whether hierarchical supervision captures biologically meaningful structure, we visualized the learned protein embeddings at the residue, domain, and lineage levels with and without hierarchical constraints. As shown in Fig.6, without hierarchical supervision, the embeddings appear disorganized, with dispersed points and poorly defined clusters. Incorporating hierarchical information yields compact clustering at the domain and lineage levels, showing that the model leverages hierarchical relationships to capture functional and evolutionary structure. This prior al-

| Method | SHS27k | | | SHS148k | | |
|---|---|---|---|---|---|---|
| | DFS | BFS | Random | DFS | BFS | Random |
| **Full** | 0.8227 | 0.8328 | 0.9115 | 0.8673 | 0.8541 | 0.9376 |
| **w/o HRC** | 0.7753 | 0.7710 | 0.9083 | 0.8630 | 0.7951 | 0.9487 |
| **w/o kw** | 0.8110 | 0.8010 | 0.9030 | 0.8591 | 0.8426 | 0.9316 |
| **w/o HRC & kw** | 0.6231 | 0.6231 | 0.8441 | 0.7509 | 0.6140 | 0.9080 |

Table 1: Performance comparison on the SHS27k and SHS148k datasets under different ablation settings. The results are reported for three sampling strategies (DFS, BFS, Random). "Full" denotes the complete model; "w/o HRC" and "w/o kw" denote the model without hierarchical attributes and keyword annotations, respectively; "w/o HRC & kw" indicates both modules are removed. Higher values indicate better performance.

lows proteins with shared characteristics to cluster more closely in the embedding space, improving discrimination between interacting and non-interacting pairs.

Moreover, we analyzed the role of non-hierarchical annotations. When non-hierarchical information was excluded and the model relied solely on hierarchical annotations, performance did not degrade, suggesting that hierarchical information is the dominant factor in guiding the learning of effective protein representations.

## 5 CONCLUSIONS

In this work, we introduce HIPPO, a hierarchical contrastive framework for cross species protein-protein interaction prediction. HIPPO integrates amino acid sequences with multi-level annotations, including biological hierarchies and protein functions, and performs structured message passing on protein-protein interaction graphs. A hierarchical supervision mechanism is proposed to enforce embeddings that reflect evolutionary and functional relationships, thereby facilitating transfer across species. Moreover, extensive experiments on diverse benchmark datasets demonstrate that HIPPO achieves consistent improvements over existing methods, particularly on challenging cross species settings. HIPPO also provides insights into protein interpretability by revealing conserved motifs and functional regions that are associated with interaction predictions. We believe that our work represents a significant step forward in cross species prediction research.

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

## A ADDITIONAL FIGURES

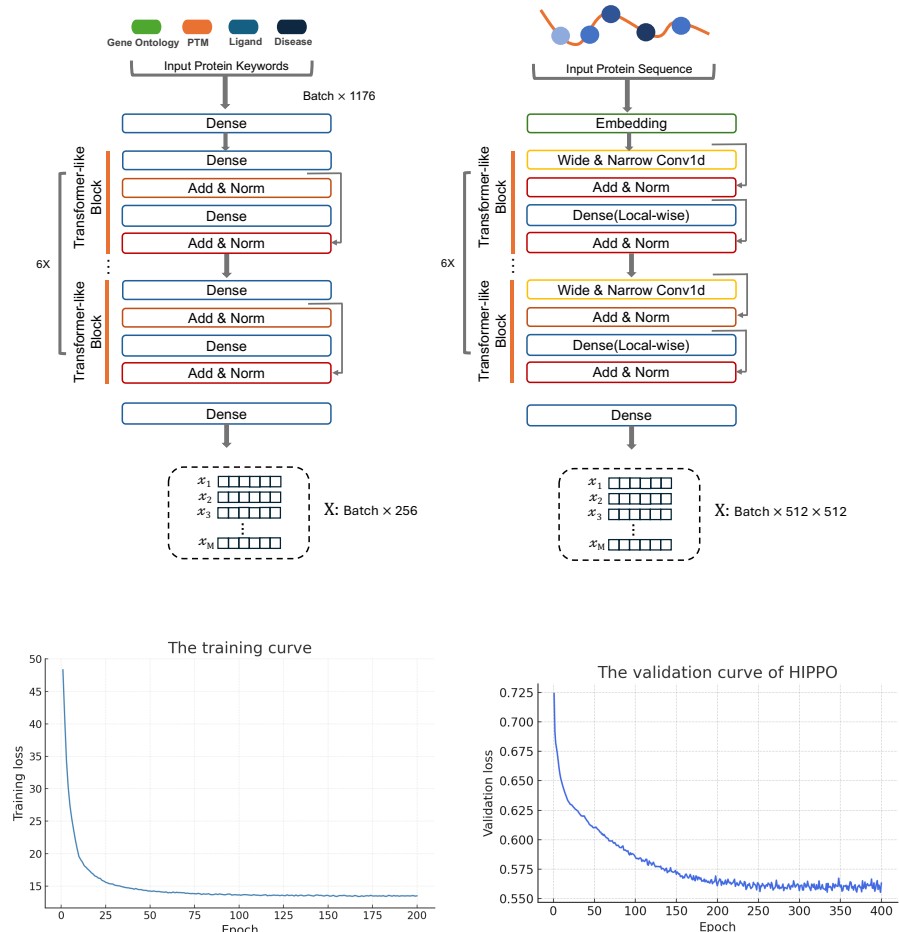

Figure A1: **Model architectures and training curves.** Overview of the sequence–annotation encoder within HIPPO. The protein sequence is embedded, processed by transformer-like blocks with 1D convolutions and local dense layers, pooled by attention to a global embedding, and projected to 256 dimensions for downstream PPI prediction.

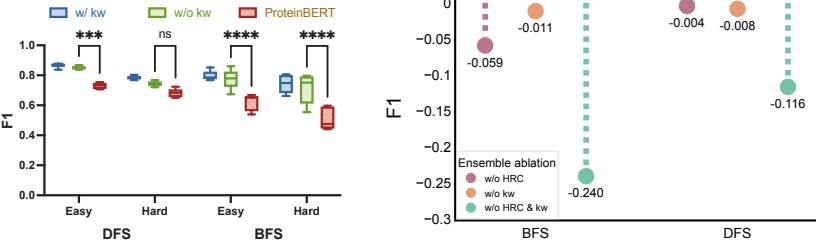

Figure A2: **Ablation and comparison on SHS148k.** Left: performance with and without keyword annotations, and ProteinBERT baseline. Significance: **** ($p < 0.0001$), *** ($p < 0.001$), ** ($p < 0.01$). Right: relative F1 drop $\Delta F_1$ when removing hierarchical attributes (HRC) and/or keyword annotations (kw).

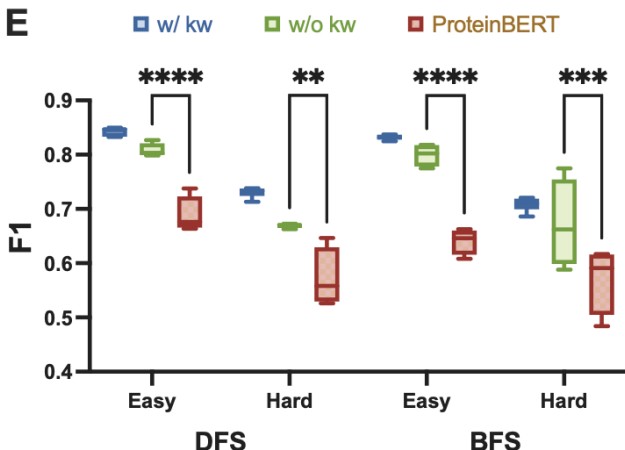

Figure A3: **Comparison of model performance on SHS27k.** Box plots show results with keyword annotations (w kw), without keyword annotations (w o kw), and the baseline ProteinBERT. Statistical significance is indicated by asterisks: **** p ¡ 0.0001, *** p ¡ 0.001, ** p ¡ 0.01.

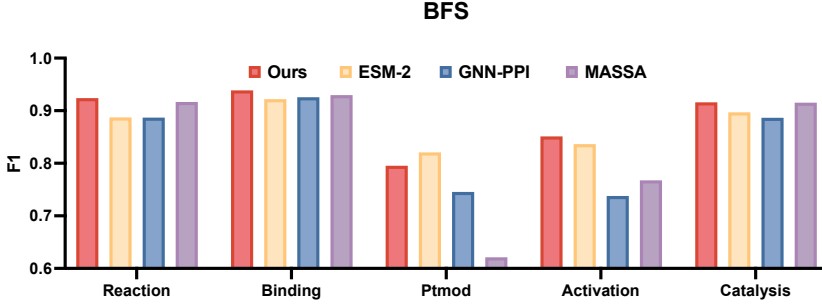

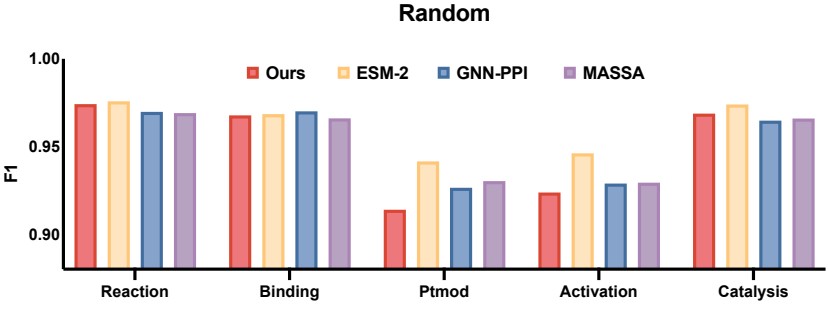

Figure A4: **Performance across PPI types on SHS27k.** F1 scores of four models (Ours, ESM2, GNNPPI, MASSA) on five interaction types (Reaction, Binding, PTM, Activation, Catalysis) under BFS (upper) and Random (lower) sampling.

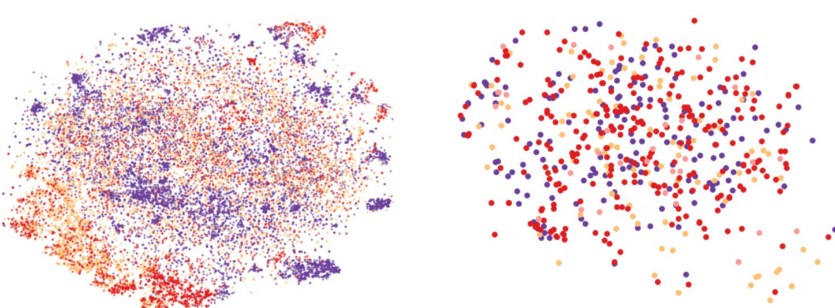

Figure A5: **Feature visualization of protein representations.** Left: domain embeddings (Protein kinase, WD40 repeats, Immunoglobulin, Ankyrin repeat). Right: lineage embeddings (Actinobacteria, Bacteroidetes, Firmicutes) on Pfam PF00144. Encoder trained without keyword pretraining; t-SNE projections shown.

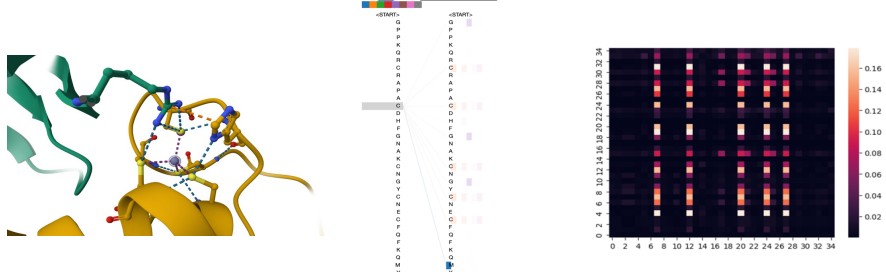

Figure A6: **Attention-based visualization of binding motifs.** Cys4 zinc-finger motif (PDB: 3VUX): structure view with zinc-coordinating residues, sequence-level attention map, and attention heatmap highlighting strong attention among binding cysteines.

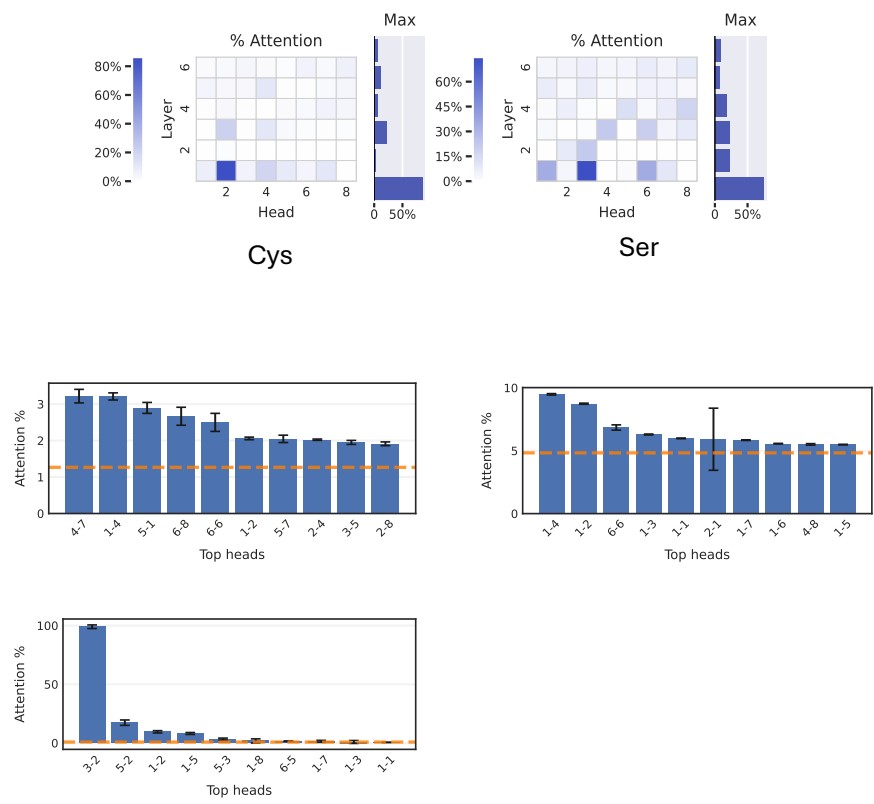

Figure A7: **Attention head specialization.** Top: per-head attention allocation to cysteine and serine across the test set. Bottom: proportion of attention on contact maps, binding sites, and PTM sites with 95% confidence intervals; dashed lines show uniform baseline.

# B    ADDITIONAL TABLES

| SHS27k | DFS | | | BFS | | |
|---|---|---|---|---|---|---|
| | All | Medium/ES | Hard/NS | All | Medium/ES | Hard/NS |
| Ours | $0.831 \pm 0.003$ | $0.843 \pm 0.003$ | $0.730 \pm 0.004$ | $0.816 \pm 0.002$ | $0.832 \pm 0.002$ | $0.708 \pm 0.005$ |
| GNN-PPI | $0.711 \pm 0.002$ | $0.722 \pm 0.002$ | $0.619 \pm 0.003$ | $0.803 \pm 0.002$ | $0.824 \pm 0.003$ | $0.688 \pm 0.007$ |
| ESM-2 | $0.723 \pm 0.005$ | $0.737 \pm 0.004$ | $0.611 \pm 0.007$ | $0.806 \pm 0.003$ | $0.823 \pm 0.003$ | $0.686 \pm 0.005$ |
| INTREPPID | $0.736 \pm 0.006$ | $0.751 \pm 0.006$ | $0.593 \pm 0.014$ | $0.722 \pm 0.022$ | $0.742 \pm 0.021$ | $0.581 \pm 0.031$ |

Table B1: **Performance on SHS27k by difficulty.** Mean $\pm$ standard deviation over repeated runs under DFS and BFS.

| SHS148k | DFS | | BFS | |
|---|---|---|---|---|
| | Medium/ES | Hard/NS | Medium/ES | Hard/NS |
| Ours | $0.866 \pm 0.006$ | $0.784 \pm 0.006$ | $0.797 \pm 0.013$ | $0.742 \pm 0.025$ |
| GNN-PPI | $0.852 \pm 0.020$ | $0.743 \pm 0.0065$ | $0.844 \pm 0.0027$ | $0.764 \pm 0.008$ |
| ESM-2 | $0.873 \pm 0.0008$ | $0.763 \pm 0.00037$ | $0.811 \pm 0.0139$ | $0.770 \pm 0.0153$ |
| INTREPPID | $0.833 \pm 0.0005$ | $0.728 \pm 0.0129$ | $0.717 \pm 0.016$ | $0.555 \pm 0.015$ |

Table B2: **Performance on SHS148k by difficulty.** Mean $\pm$ standard deviation under DFS and BFS.

| Method | Reaction | Binding | PTM | Activation | Inhibition | Catalysis | Expression |
|---|---|---|---|---|---|---|---|
| Ours | 0.8606 | 0.8811 | 0.8899 | 0.7737 | 0.6304 | 0.9131 | 0.2209 |
| ESM-2 | 0.6514 | 0.7957 | 0.6905 | 0.6925 | 0.7673 | 0.8412 | 0.3780 |
| GNN-PPI | 0.7676 | 0.8142 | 0.6616 | 0.7165 | 0.7779 | 0.8551 | 0.3515 |
| MASSA | 0.7134 | 0.8085 | 0.6851 | 0.7461 | 0.7338 | 0.8667 | 0.3771 |

Table B3: **F1 across interaction categories on SHS27k.**

| Method | Reaction | Binding | PTM | Activation | Inhibition | Catalysis | Expression |
|---|---|---|---|---|---|---|---|
| Ours | 0.9239 | 0.9387 | 0.7954 | 0.8510 | 0.7513 | 0.9159 | 0.5054 |
| ESM-2 | 0.8871 | 0.9219 | 0.8208 | 0.8365 | 0.7308 | 0.8969 | 0.4424 |
| GNN-PPI | 0.8868 | 0.9258 | 0.7455 | 0.7377 | 0.7661 | 0.8864 | 0.4884 |
| MASSA | 0.9164 | 0.9292 | 0.6209 | 0.7673 | 0.7355 | 0.9149 | 0.4217 |

Table B4: **F1 across interaction categories on SHS148k.**

| Species | E. coli | Yeast | C. elegans | Arabidopsis | Mouse | D. melanogaster |
|---|---|---|---|---|---|---|
| Pairs count | 73224 | 769028 | 3123146 | 4251642 | 4850272 | 1495180 |

Table B5: **Cross-species PPI pair counts.**

| Method | E. coli | Yeast | C. elegans | Arabidopsis | Mouse | D. melanogaster |
|---|---|---|---|---|---|---|
| Ours | 0.7559 | 0.7452 | 0.4940 | 0.7028 | 0.7177 | 0.5108 |
| GNN-PPI | 0.2671 | 0.1968 | 0.3831 | 0.3511 | 0.3942 | 0.4709 |
| ProteinBERT | 0.2899 | 0.2182 | 0.2626 | 0.3638 | 0.4776 | 0.0600 |
| ESM-2 | 0.4067 | 0.5079 | 0.4916 | 0.2371 | 0.2851 | 0.2598 |
| INTREPPID | 0.4586 | 0.4264 | 0.3533 | 0.2951 | 0.5079 | 0.1423 |
| PIPR | 0.3898 | 0.4337 | 0.4833 | 0.4509 | 0.4408 | 0.4374 |

Table B6: **Cross-species F1 scores across six organisms.**

## C ADDITIONAL METHODS

### C.1 DATA PREPARATION AND ANALYSIS

During the pretraining stage, we employed the Swiss-Prot dataset from UniProtKB Bairoch & Apweiler (2000) due to its extensive protein data coverage and high-quality, manually curated annotations. Protein attributes include various protein families and clans accessible through the Pfam database Mistry et al. (2021), and annotations curated within the UniProtKB Keywords section. The keywords section incorporates controlled vocabulary terms manually annotated to include Gene Ontology (GO) terms, disease associations, protein domains, ligands, and post-translational modifications (PTMs) The UniProt Consortium (2025).

During protein–protein interaction (PPI) prediction stage, we utilized three datasets derived from STRING Szklarczyk et al. (2021; 2025): SHS148k and SHS27k. SHS27k and SHS148k datasets were created by selecting proteins longer than 50 amino acids with less than 40% sequence identity to form more challenging subsets. SHS27k comprises 63,408 interactions among 1,690 proteins, while SHS148k includes 36,902 interactions among 5,189 proteins Lv et al. (2021). Three partition methods—Random, Breadth-First Search (BFS), and Depth-First Search (DFS)—were employed for dataset splitting Lv et al. (2021); Hu et al. (2023). To rigorously assess model generalization, we evaluated performance on unfamiliar proteins within the test set by checking if interacting proteins existed in the training set, with detailed splitting procedures described in the Results section.

Hierarchical protein labels represent multiple interconnected annotations related to protein functions and residue-level details, organized in a tree structure. In this hierarchy, leaf nodes correspond to unique sequence identifiers, while non-leaf nodes indicate evolutionary classifications at various hierarchical levels. Higher hierarchy levels (e.g., Clan $l_1$) represent broader evolutionary relationships, situated closer to the root, whereas lower levels (e.g., Family $l_2$) represent narrower classifications. Clans integrate multiple related protein domain families, reflecting extensive evolutionary connections and similarities. Positive sequence pairs at a given hierarchical level $l \in L$ are defined as sequences sharing common ancestry up to level $l$ but diverging thereafter. As illustrated in Fig. 1, a pair at the clan level $l_1$ implies their lowest common ancestor is at this hierarchical level. Our framework comprises 6,329 distinct families and 621 clans, collectively forming a hierarchical clan-family tree capturing intrinsic protein properties such as evolutionary relationships, sequence similarities, and structural homologies Paysan-Lafosse et al. (2025).

### C.2 TRAINING DETAILS

For the pretraining stage on Swiss-Prot, we used a batch size of 128 and trained the model for up to 100 epochs. The initial learning rate was set to $3 \times 10^{-4}$ with cosine decay, a minimum learning rate of $1 \times 10^{-6}$, and a warmup schedule with 3,000 steps (warmup learning rate $1 \times 10^{-6}$). We applied weight decay of 0.05. The queue size for contrastive learning was set to 65,536 and the momentum coefficient $\alpha$ was 0.4. A learning rate decay rate of 0.9 was applied across epochs. For supervised PPI prediction, we trained with a batch size of 128 for 400 epochs. The learning rate was set to $1 \times 10^{-6}$. Checkpoints were saved throughout training to monitor performance.

### C.3 CROSS-SPECIES PPI PREDICTION

We train on human PPIs (SHS148k) and evaluate on six species: *Escherichia coli*, yeast, *Caenorhabditis elegans*, *Arabidopsis thaliana*, mouse, and *Drosophila melanogaster*. Test PPIs are disjoint from training. Performance is measured by F1; counts and results are in Tables B5 and B6. Our hierarchical model achieves the best F1 on all species, with higher scores on evolutionarily closer organisms (e.g., mouse).

### C.4 ATTENTION-BASED BINDING SITE PREDICTION

Given an input sequence of length $L$, the model produces self-attention matrices $A_{l,h} \in \mathbb{R}^{L \times L}$ for layer $l$ and head $h$. We aggregate attention over selected layers and heads and score residue $i$ by

$$a_i = \frac{1}{N_h N_l} \sum_{h=1}^{N_h} \sum_{l=1}^{N_l} \sum_{j=1}^{L} A_{l,h}[i,j].$$

Top-$k$ residues by $a_i$ are predicted as binding sites; consensus voting across heads improves robustness. Overlap and false-positive rates are reported; qualitative cases in Figure **??** show that HRC pretraining improves specificity and accuracy.

### C.5 ATTENTION ANALYSIS

We visualize residue–residue attention (Figure A6) and quantify per-head specialization (Figure A7). For each head, we compute the fraction of attention allocated to specific residue types (e.g., cysteine, serine) and to annotated functional positions (contacts, binding, PTMs), normalized by total head attention. Heads showing large fractions above background indicate biologically meaningful specialization.

