# OpenReview forum: "Cross-Species Protein Interaction Prediction by Hierarchical Contrastive Pretraining"
_ICLR.cc/2026/Conference — ICLR 2026 Conference Withdrawn Submission_

### Official Review · Reviewer_oF8H · 2025-10-31

**Soundness:** 2
**Presentation:** 1
**Contribution:** 3
**Rating:** 4
**Confidence:** 4

**Summary:**

This work introduces HIPPO, a hierarchical contrastive framework that performs cross-species PPI prediction.

HIPPO utilizes a hierarchical contrastive pretraining strategy to align protein sequence representations with function annotations (e.g., UniProt keywords), and biological hierarchies (Pfam families and clans). A graph neural network is then used for PPI prediction using data from the STRING database. On benchmarks, HIPPO outperforms baselines. Interpretability analyses are also performed to look for conserved motifs and functional regions associated with interaction predictions.

**Strengths:**

1. Challenging setting of cross-species PPI is important but understudied. It’s nice that even for the intra-species setting, the authors had used DFS and BFS splits to control for test-train similarity.
2. Ablations are illuminating for future research, especially the fact that without hierarchical contrastive learning and keyword annotations, performance will drop by ~10-20%
3. Interpretability analyses are a good contribution for understanding why things work, beyond pure metrics-based comparisons, and good to encourage for works that will follow.

On the whole, I think the problem space and the results are pretty good (especially considering the difficulty of the test/train splits), but the presentation as it currently stands is too under-polished for publication.

**Weaknesses:**

1. Pfam and Swiss-Prot annotation quality and availability will bottleneck the performance.
2. According to ablations, the keyword annotations doesn’t seem as important as the hierarchical contrastive approach.
3. Presentation shortcomings are critical. Without these being cleaned up, it would be difficult to accept this paper. It generally feels like this is a partially complete paper. Mistakes include but are not limited to:

* “Based on xxxx.” line 048.
* Some references unresolved:  line 053, 084, etc.
* Fig 5's caption is "Sample figure caption"
* Content in Fig 3 and Fig 5AB is somewhat redundant? Can they be in the same panel instead?
* Figure 1 caption is not clear. Makes it hard to fully grasp what’s happening in the work.
* Figure 6: caption says (A) and (B), but the figure doesn’t have a B panel. I also don’t see what the “**”, “***” in the caption refers to.
* Nitpick: Table 1 formatting does not follow ICLR table styles.
* Hierarchical Multi-label Constraint Enforcing Contrastive Loss reference is important for the methodology, but is unresolved
* Some references in the bibliography are duplicated.

**Questions:**

1. Why exactly do we need to do hierarchical contrastive learning? Is it actually more effective than just normal contrastive learning? Is it possible to still include any results related to this?
2. Could authors provide any intuition for why keyword annotations could help with PPI specifically?
3. I might’ve missed these details, but:
* can authors clarify what we do with the Swiss-Prot sequences that don’t have corresponding labels in GO and Pfam hierarchies? Is the 440K before or after filtering for the intersection?
* Are the encoders frozen during PPI training, or are they fine-tuned end-to-end with the GIN?
* Is the ALM trained from scratch or preinitialized? What about the PLM?

---

### Official Review · Reviewer_HF4a · 2025-11-01

**Soundness:** 2
**Presentation:** 1
**Contribution:** 2
**Rating:** 4
**Confidence:** 4

**Summary:**

The paper presents HIPPO (Hierachical Contrastive Pretraining), a new framework for PPI prediction. HIPPO works by integrating multiple data types, such as amino acid sequences, functional annotations, and biological hierarchies (like protein families and clans) into a unified representation. By using hierarchical contrastive learning, the model learns to align proteins not just by their sequence but also by their functional and evolutionary relationships. Experiments show that HIPPO achieves state-of-the-art performance, significantly outperforming existing methods in both intra-species and, crucially, cross-species prediction tasks. The model is also shown to be interpretable, as its attention mechanisms can identify biologically significant regions like conserved motifs and binding sites.

**Strengths:**

1. This work is a  case study in the value of integrating explicit domain knowledge into deep learning models. It shows that while large-scale pretraining on sequence alone (like in ESM2) is powerful, performance can be significantly enhanced by also supervising the model with structured biological priors
2. The authors provide clear quantitative and qualitative evidence for their central claim. The t-SNE visualizations in Figure 6 compellingly demonstrate that adding hierarchical supervision results in a much more organized and biologically-structured embedding space at the domain and lineage levels, compared to the disorganized space from the model trained without it.

**Weaknesses:**

1. The paper's innovation is limited. The methodology combines several standard techniques, including a symmetric InfoNCE loss for multimodal alignment ( $\mathcal{L_{SAC}}$ ), and a focal loss for sequence-annotation matching ( $\mathcal{L}_{SAM}$ ). The concept of aligning sequence and annotation embeddings is also a core principle of prior work like ProteinBERT.

**Questions:**

1.  Could you provide results for the full model without $\mathcal{L_{SAC}}$ and the full model without $\mathcal{L_{SAM}}$? This would help justify the inclusion of all three loss components and demonstrate that $\mathcal{L_{SAC}}$ and $\mathcal{L_{SAM}}$ are capturing complementary, non-redundant signals.

---

### Official Review · Reviewer_MFYy · 2025-11-01

**Soundness:** 3
**Presentation:** 3
**Contribution:** 2
**Rating:** 4
**Confidence:** 4

**Summary:**

The authors proposes HIPPO, a framework for cross-species protein–protein interaction (PPI) prediction with Hierarchical contrastive pretraining (HiMulConE) enforcing consistency across biological hierarchies (clans, families, domains), Sequence–annotation contrastive alignment, Graph-based downstream PPI classification via a GIN encoder. Results shows out-of-distribution generalization across species by capturing multi-level protein structure–function relationships. It is evaluated on intra-species (SHS27k, SHS148k) and cross-species (6 organisms) datasets, reportedly achieving state-of-the-art F1 scores and improved interpretability.

**Strengths:**

The main strength of this paper is its strong biological motivation and the deployment of the hierarchical pretraining scheme

1. The focus on cross-species PPI stands out because it's often hard to resolve through homology based methods. Learning transferable structured protein embeddings is clearly motivated.
2. Strong empirical evaluation over multiple datasets and thoughtful dataset split.
3. good interpretability demonstrating attention based visualization with relevant biological insights.

**Weaknesses:**

The main weakness of this paper is methodological clarity and the articulation of the proposed method integration strategy.

1. The core methods—contrastive learning, InfoNCE, and sequence–annotation alignment—are directly borrowed from CLIP-style or SimCLR formulations. What is fundamentally new in HIPPO’s algorithm beyond integrating existing components (contrastive loss + GIN encoder + hierarchy-aware sampling)? Analysis of the quality of alignment will be helpful to justify the method development.

2. In the hierarchical contrastive loss, does this loss enforces the transitive hierarchy? Comparison with other contrastive objectives?

3.The choice of a GIN as the graph encoder is not well explained.

4. The interpretability analysis is a good addition for model analysis but it mostly feel adenoidal, further quantitative analysis such as overlap percentage with binding sites will help.

**Questions:**

1. in the symmetric InfoNCE and focal-loss SAM objective, what's the motivation behind this choice? Why not a single multimodal contrastive loss like CLIP?

2. It’s unclear how “annotation embeddings” are learned — is the “annotation language model” pre-trained on GO terms or jointly optimized?

3. How were Pfam clans/families selected for hierarchical supervision, and did you evaluate the effect of hierarchy depth (number of levels)?

4. How does HIPPO handle proteins missing complete hierarchical annotations?

---

### Official Review · Reviewer_rXya · 2025-11-01

**Soundness:** 2
**Presentation:** 1
**Contribution:** 2
**Rating:** 2
**Confidence:** 4

**Summary:**

This paper proposes HIPPO, a hierarchical contrastive framework for cross-species protein-protein interaction prediction, addressing the limitations of poor intra-species accuracy and weak cross-species generalization of existing methods. HIPPO integrates amino acid sequences, biological hierarchies and functional annotations into a unified representation learning objective via hierarchical contrastive learning and multimodal sequence-annotation alignment, and uses graph neural networks to capture global PPI network context.

**Strengths:**

1. HIPPO integrates amino acid sequences, biological hierarchies and functional annotations, enabling more comprehensive protein representation than sequence-only or graph-only baselines.
2. Attention-based interpretability reveals biologically meaningful features, enhancing prediction reliability.

**Weaknesses:**

1. The paper contains placeholders and editing errors (e.g., Line 48 “Based on xxxx.” Line 53 and Line 84 “?”).
2. The proposed HIPPO builds on existing contrastive and multimodal frameworks, offering only incremental improvements without clear innovation.
3. The experiments rely on outdated models; stronger, more recent baselines should be included to validate the method’s effectiveness.
4. The framework diagram of HIPPO (Figure 1) is rough. The author should add more details to enhance its readability.

**Questions:**

The questions are listed under Weaknesses above.

---

### Note · Authors · 2025-11-26

**Comment:**

After discussion with collaborators, we believe the work is better suited for a different venue that aligns more closely with our technical contributions. We would like to withdraw the submission. We thank the reviewers and area chairs for their time and constructive comments.

**Withdrawal Confirmation:**

I have read and agree with the venue's withdrawal policy on behalf of myself and my co-authors.